# Flavonoid Derivatives as Potential Cholinesterase Inhibitors in Scopolamine-Induced Amnesic Mice: An In Vitro, In Vivo and Integrated Computational Approach

**DOI:** 10.3390/brainsci12060731

**Published:** 2022-06-02

**Authors:** Fakhria A. Al-Joufi, Syed Wadood Ali Shah, Mohammad Shoaib, Mehreen Ghias, Atif Ali Khan Khalil, Syed Babar Jamal, Syed Muhammad Hassan Shah, Muhammad Zahoor

**Affiliations:** 1Department of Pharmacology, College of Pharmacy, Jouf University, Skaka 72341, Aljouf, Saudi Arabia; faaljoufi@ju.edu.sa; 2Department of Pharmacy, University of Malakand, Dir (Lower), Chakdara 18800, Khyber Pakhtunkhwa, Pakistan; pharmacistsyed@gmail.com (S.W.A.S.); mohammadshoaib13@yahoo.com (M.S.); mehreenghias@yahoo.com (M.G.); shafi_ullah34@yahoo.com (S.); 3Department of Biological Sciences, National University of Medical Sciences, Rawalpindi 46000, Punjab, Pakistan; atif.ali@numspak.edu.pk (A.A.K.K.); babar.jamal@numspak.edu.pk (S.B.J.); 4Department of Pharmacy, Sarhad University of Science and Information Technology, Peshawar 25000, Khyber Pakhtunkhwa, Pakistan; syedhassan41@gmail.com; 5Department of Biochemistry, University of Malakand, Dir (Lower), Chakdara 18800, Khyber Pakhtunkhwa, Pakistan

**Keywords:** flavones, enzyme inhibition, docking, Alzheimer’s disease, AChE, nootropic agents

## Abstract

Flavonoids are one of the most exciting types of phenolic compounds with a wide range of bioactive benefits. A series of flavone derivatives (**F1**–**F5**) were previously synthesized from substituted O-hydroxy acetophenone and substituted chloro-benzaldehydes. The titled compounds **F1**–**F5** in the present study were evaluated for their anticholinesterase potential (against AChE and BuChE). The obtained results were then validated through a molecular docking approach. Compound **F5** was found to be the most potent inhibitor of AChE (IC_50_ = 98.42 ± 0.97 µg/mL) followed by compound **F4**, whereas compound **F2** was found to be the most promising inhibitor of BuChE (IC_50_ = 105.20 ± 1.43 µg/mL) among the tested compounds. The molecular docking analysis revealed a similar trend in the binding affinity of compounds with the targeted enzymes and found them to be capable of forming highly stable complexes with both receptors. The selected compounds were further subjected to in vivo assessment of cognitive function in a scopolamine-induced amnesic animal model, in which almost all compounds **F1**–**F5** significantly attenuated the amnesic effects as evaluated through Y-Maze Paradigm and novel object discrimination (NOD) tasks, findings that were further supported by ex vivo experimental results. Among (**F1**–**F5**), **F5** showed significant anti-amnesic effects in scopolamine-induced amnesic models and ameliorated the memory loss in behavioral model studies as compared to counterparts. In ex vivo study, noteworthy protection from oxidative stress in the brains of scopolamine-induced amnesic mice was also recorded for **F5**. These findings also confirmed that there were no significant differences among the in vivo and ex vivo results after administration of **F1**–**F5** (7.5 or 15 mg/kg) or donepezil (2 mg/kg). These synthesized flavonoids could serve as potential candidates for new neuroprotective and nootropic drugs. However, further studies are needed to validate their observed potential in other animal models as well.

## 1. Introduction

Humans are the super creature of this planet Earth, capable of learning and memorizing things. It has been claimed for a long time that acetylcholine (ACh) has an imperative role in cognitive functions, mainly in memory and learning [1,2]. Research studies on patients with Alzheimer’s and experimental manipulation in animals (cholinergic lesions and drug studies) strengthened the idea of ACh involvement in the memory and learning process. Several experimental reports showed learning and memory impairments using various methodologies and concluded that ACh was associated with memory functions. Most importantly, short-term and working memory were prominently affected [3].

Normal cholinergic activity involves the sequential release, binding and deactivation of ACh (a principal neurotransmitter) by an enzymatic acetyl-cholinesterase (AChE), while at the synapses, insufficiency or shortage in cholinergic transmissions results in abnormal cholinergic activity, which may be due to reduction in acetylcholine (ACh) production or its excess hydrolysis/deactivation by AChE [4,5]. In the brain, after the use of acetylcholine, its breakdown occurs via the acetylcholinesterase enzyme. Therefore, the use of a cholinesterase inhibitor is one of the treatment approaches for AD that will keep acetylcholine concentrations high by reducing the activity of the acetylcholinesterase enzyme. Butyrylcholinesterase (BuChE) also causes the inactivation of acetylcholine (ACh) neurotransmitters and can be targeted therapeutically in Alzheimer’s disease [6].

Currently, commercially natural and synthetic acetylcholinesterase inhibitors are available in the market for the treatment of AD. The majority of available products are cholinesterase inhibitors such as galantamine, rivastigmine and donepezil, while some are NMDA receptor antagonists, such as memantine and rivastigmine, which are natural compounds. Galantamine is naturally obtained from *Galanthus woronowii*, *Galanthus nivalis*, and some other plants in the *Amaryllidaceae* family, while donepezil is of synthetic origin. The AChE inhibitors promote the dynamics of ACh by restraining the activity of the enzyme (AChE), thus increasing the availability and interaction time with cholinergic receptors of Ach [7,8]. In the present study, we evaluated synthetic flavonoids **F1**–**F5** as acetylcholinesterase and butyrylcholinesterase inhibitors using in vitro, in vivo, and ex vivo examination and integrated computational approaches.

## 2. Materials and Methods

### 2.1. Chemicals and Equipment

AChE (electric eel type-VI-S), BuChE (equine serum lyophilized), acetylthiocholine and butyryl thiocholine iodide, DTNB, donepezil and galantamine hydrobromide lycoris Sp. were purchased from Sigma-Aldrich (Taufkirchen, Germany). Synthetic flavonoids **F1**–**F5** previously synthesized by our group were used in the study [9,10,11].

### 2.2. In-Vitro Anticholinesterase Activity

For the determination of acetylcholinesterase inhibitory potentials, AChE and BuChE were used. Briefly, flavone derivatives (**F1**–**F5**) of various concentrations (50 µL) and AChE (0.5 mL) were mixed in a test tube, and the tube was set on the incubator (25 °C). To the tube were added DTNB (100 µL) and buffer (2.4 mL). The tube was incubated at 25 °C for 5 min as pre-incubation. The reaction was started by adding ATChI (40 µL), and the mixture was incubated at 25 °C for 20 min. The absorbance at 412 nm was measured spectrophotometrically in triplicate [9]. Similarly, the butyrylcholinesterase inhibitory potentials of flavone derivatives (**F1**–**F5**) of various concentrations were assessed using the BuChE enzyme, DTNB and BTChI, and the absorbance was measured at 412 nm. Donepezil was taken as standard. The data were recorded in triplicate, and IC_50_ was calculated [12].

### 2.3. Receptor Preparation

We utilized the PDB structures of the human acetylcholinesterase complex with donepezil (PDB ID: 4EY7) and butyrylcholinesterase complex with tacrine (PDB ID: 4BDS) with a resolution of 2.35 and 2.10 Å, respectively. For both, the proteins missing residues were included through MOE software (Montreal, Canada). All water molecules were deleted except the conserved water molecules. Energy minimization was also performed for both of the complex structures.

### 2.4. Re-Docking Setup and Ligand Preparation

In order to validate the docking software, redocking was performed. The co-crystallized ligand was docked into the binding sites of acetylcholinesterase (AChE, PDB ID: 4EY7) and butyrylcholinesterase (BChE, PDB ID: 4BDS). Each re-docked pose was assessed through RMSD values [13].

Three-dimensional structures of flavones (**F1**–**F5**) were sketched through MOE software. The MMFF94 force field was applied on the ligands, and energy minimization was performed through MOE software.

### 2.5. Docking of Acetyl and Butyryl Cholinesterase Inhibitors

For obtaining structural features involved in the binding mode of flavones, docking was performed. It obtained the suitable conformation of the ligand in the active site of the receptor and utilized the scoring function to satisfactorily define the best pose of the ligand. **F1**–**F5** were docked into the active sites of acetylcholinesterase (PDB ID: 4EY7) and butyrylcholinesterase (PDB ID: 4BDS) by using MOE software. For each compound, 30 poses were generated. Finally, the interaction pattern and scoring function were utilized to choose the best conformation of the compound.

### 2.6. Dynamics Understanding and Binding Free Energies of Systems

Molecular dynamics study of the protein–ligand complexes was accomplished using Amber16 software [14]. Protein processing was performed using ff14SB force field [15], while a ligand topology file was prepared and parameterized through Amber general force field (GAFF) [16]. Solvent water molecules were added in a cubic box where the distance between the complex and box boundary was set to 12 Å. An appropriate number of Na^+^ ions were used to neutralize each system. System energy minimization was achieved by subjecting each system to 1500 steps of steepest descent and conjugate gradient steps. Heating was conducted for 50 picoseconds where temperature was scaled to 300 K gradually with restraint applied on carbon alpha. Next, an equilibration step was performed with position restraints. The SHAKE algorithm [17] was applied to each system to provide constraint on hydrogen bonds, keeping a defined cut-off distance. A production run of 5000 picoseconds was carried out in the presence of NVT ensemble.

Furthermore, estimation of binding free energies was completed as per MMPBSA.py protocol [18]. Net binding energy was determined for complex, receptor and ligands separately and consequently underwent total binding free energy change estimation for the systems. In total, 100 frames were extracted from simulation trajectories and analyzed by the MMPBSA pipeline.

### 2.7. Animals

A total of 134 healthy Balb/C mice aged 8–10 weeks old (19–23 g) were procured from the Veterinary Research Institute (VRI), Lahore and kept in an animal house in standard plastic cages under standard laboratory conditions with 25 ± 2 °C temperature, relative humidity of 55–65% and 12 h light/12 h dark cycle with a standard diet and water ad libitum. Two weeks before the experiment, the animals were acclimatized to laboratory conditions. The animals were treated following the protocols mentioned in the “Animals Byelaws 2008 of University of Malakand (Scientific Procedures Issue-I)”. Approval for the study was granted by the Ethical Committee of the Department of Pharmacy, in accordance with the Animals Byelaws 2008 of University of Malakand, vide notification no: Pharm/EC-SyFl/11-22/21.

### 2.8. Acute Toxicity Study

For assessment of acute toxicity of the flavones (**F1**–**F5**), animals (21 in number) were divided into groups of 3 animals each. The reported protocols by Lorke were followed with slight modification to perform tests in two phases. In the first phase, one (control) group of animals were given Tween-80 (2%), and oral doses of **F1**–**F5** were given to the remaining groups, respectively, at various doses (mg/kg body weight). During the 2nd phase, respective oral doses of **F1**–**F5** were given, and the animals were observed for 24 h for any kind of physical or behavioral change, followed by careful observation for two weeks to assess any kind of physical or behavioral change. The animals were housed in plastic cages [19]. After two weeks, the animals were sacrificed by euthanasia with isoflurane in a humane manner for assessment of the effects on total weight and weight of the vital organs. The biochemical parameters of blood and histological studies of vital organs (kidney and liver tissue) were also assessed. Selection of doses for in vivo pharmacological assessment of cognitive function using the animal model was carried out from in vivo toxicological studies as per OECD, 2001 guidelines and the approach to practical acute toxicity testing by Dietrich Lorke (1983) and ARRIVE guidelines. With the toxicity data at hand, effective doses (mg/kg b.w.) were selected for behavioral studies after preliminary pharmacological assessment in our laboratory. The preliminary pharmacological activity was assessed at various dose concentrations of **F1**–**F5** (2.5–20 mg/kg b.w.) to determine the effective dose for assessment of cognitive function using animal models. The findings on preliminary pharmacological activity aided in standardizing the **F1**–**F5** for assessment and selection of doses for pharmacological investigation.

### 2.9. Experimental Design for Anti-Amnesic Activity in Scopolamine-Induced Amnesic Model

The animals were divided randomly into experimental groups (*n* = 8), consisting of normal control, amnesic control (scopolamine-treated), amnesic mice treated with 7.5 and 15 mg/kg b.w. of **F1**–**F5,** and a donepezil (2 mg/kg) positive control group. The animals in the control (normal) and scopolamine amnesic groups were administered vehicle (Tween-80, 2%) only. Treated animals received **F1**–**F5** suspended in Tween-80 (2%) p.o. at their respective doses for a period of 4 weeks. The positive control group was treated with standard drug donepezil (2 mg/kg). For assessing behavioral effects using the Y-Maze Paradigm and novel object discrimination (NOD) task, scopolamine (1 mg/kg, i.p.) was given to the different groups, 30 min after the respective treatments, to induce memory impairment (amnesia) in mice. All of the tests were performed between 8:00 a.m. and 6:00 p.m. to avoid sham results in the performance of the animals.

### 2.10. Assessment of Cognitive Function

Behavioral tasks on the Y-Maze Paradigm and “NOD” tasks were conducted in the 5th week of study to determine learning and memory functions. The apparatus arena was cleaned with 70% ethanol during the inter-trial interval to prevent a confounding error due to the influence of odor [20,21].

#### 2.10.1. Y-Maze Paradigm

The Y-Maze task is a non-invasive and reliable behavioral test that was used for memory assessment of spontaneous alternation for **F1**–**F5** based on previously reported protocols. A single session of Y-Maze was used to record the spontaneous alternation behavior of the animals to determine exploratory behavior and instant memory functioning. The apparatus was made of three arms in a “Y” shape (35 cm × 8 cm × 15 cm) with an equilateral triangular central area. In brief, each of the mice, previously naive to the maze, was placed at the terminal end of one arm. The animals were allowed to move and explore freely for 8 min through the maze. The number of arm entries was noted. One arm entry was considered when the hind paws of the animal were completely placed inside the arm, and the series of arm entries were documented. The spontaneous alternation in percent was calculated.

#### 2.10.2. NOD Task

Animals were challenged for the novel object discrimination task for **F1**–**F5** as per previously reported protocols. After habituation and acclimatization, animals in each group had two consecutive object exploration trials (5 min each) for this test, with a break of 4 h in between two trials. In the familiarization phase (sample), the animals were challenged to explore two objects similar to each other. In the second phase (test), any of the two objects was changed by a new object. Exploration time in sec for the objects, including chewing, licking, sniffing or pointing the vibrissae of nose towards the object, was recorded. The discrimination ratio (%DI) was then determined using:(T novel object − T familiar object)/(T novel object + T familiar object) × 100(1)

### 2.11. Measurement of Antioxidant Enzyme Activities and Oxidative Stress Markers

The brain was extracted, and a homogenate (10% *w*/*v*) in 0.1 M of phosphate buffer (pH 7.4) was prepared by centrifugation for the assessment of brain oxidative status and estimating brain acetylcholinesterase activity. Catalase (CAT), superoxide dismutase (SOD), glutathione (GSH) and level of lipid peroxidation (malondialdehyde, MDA) were quantified as oxidative stress markers. Acetylcholinesterase activities were quantified in homogenate as reported in the method illustrated by Ellman et al. (1961) [22].

### 2.12. Statistical Analysis

The data obtained were expressed as mean ± SEM and were statistically analyzed with GraphPad Prism 5 version 5.01 using analysis of variance followed by Dunnett’s post hoc multiple comparison test. All group data were considered significant at *p* < 0.05.

## 3. Results

### 3.1. In Vitro Anticholinesterase Activity

The structure of flavone derivatives and in vitro anticholinesterase inhibitory potential capacity of the flavone derivatives were determined, and IC_50_ values are given in Table 1.

Notably, the halogenated flavones such as **F3**, **F4**, and **F5** possessed potent activity as compared to the other derivatives of flavones (**F1** and **F2**). From the current results, it was suggested that the potency of the individual flavones may increase or decrease with the addition of moiety or changes in their position. From the results in Table 1 and Table 2, it is clear that **F5** demonstrated inhibitory activity against AChE (IC_50_ = 98.42 ± 0.97 µg/mL) to a greater extent than the rest of the flavonoids, (**F1** to **F4)**. Moderate inhibition of AChE activity was demonstrated by **F4** (IC_50_ = 112.33 ± 1.16 µg/mL). Weak inhibitory action on AChE was observed with **F3** (IC_50_ = 126.29 ± 1.33 µg/mL), **F2** (IC_50_ = 131.33 ± 1.12 µg/mL) and **F1** (IC_50_ = 165.21 ± 1.53 µg/mL). Similarly, the results for BuChE inhibitory activity showed that **F2** had more promising activity against BuChE (IC_50_ = 105.20 ± 1.43 µg/mL) than the rest of the flavonoids (**F1, F3** to **F5)**. Standard donepezil possessed potent responses against both enzymes with IC_50_ = 4.91 ± 0.51 µg/mL and IC_50_ = 3.98 ± 0.67 µg/mL, respectively.

### 3.2. Validation of Docking Protocol

Redocked pose of AChE was found at a similar position when compared to reference ligand with a root mean square deviation (RMSD) value of 0.33 Å (Figure 1a,b). Similarly, in the case of butyrylcholinesterase complex, re-docked pose was also found at the same position when compared to the reference ligand with an RMSD of 0.22 Å (Figure 2a,b). Similar interaction patterns were observed among the co-crystallized ligand and the re-docked pose. Ultimately, MOE software was used for the docking of flavone compounds.

### 3.3. Docking of Flavones against Acetylcholinesterase

The binding patterns of synthesized flavones (**F1**–**F5**) were determined through a docking experiment. The most active flavone, compound **F5** (Figure 1a), established a hydrophobic interaction between 4-pyranone and Trp86 of AChE and a halogen bond between chlorophenyl and Ser203. The additional non-covalent halogen bond interaction was present in compound **F5,** validating its high experimental activity.

Compound **F2** (Figure 1b), the least active flavone, lacked the hydrophobic interaction present in compound **F5** and contained a halogen bond between chlorophenyl and Ala127. Docking interactions of compound **F2** validated its lower experimental activity. Hydrophobic interaction was observed in compound **F1** between the phenyl attached to 4-pyranone and Gly121.

Compound **F3** formed a pi-cation interaction between the phenyl attached to 4-pyranone and Tyr124, whereas compound **F4** formed a hydrophobic interaction between the phenyl group attached to 4-pyranone and Gly121 of AChE. The compounds **F1** and **F4** showed similar interactions with Gly121, validating the comparative experimental inhibitory activities.

### 3.4. Docking of Flavones against Butyrylcholinesterase

Compound **F2**, the most active flavone (Figure 2a), formed a pi-cation interaction between 4-pyranone and water molecules, and established two halogen bonds between the chlorophenyl and water molecules. The least active flavone compound **F3** (Figure 2b) formed only a pi-cation interaction between 4-pyranone and water molecule, validating the difference in biological activities of the most and least active compounds. In the case of compound **F4,** pi-cation interaction among chlorophenyl and water molecules was observed; contrarily, compound **F5** established two pi-cation interactions between phenyl attached to 4-pyranone and Trp231 of BChE. The interaction patterns of compounds **F2** and **F5** confirmed their relative biological activities (IC_50_ μg/mL). Compound **F1** formed one pi-cation interaction between 4-pyranone and two hydrogen bonds between the carbonyl of 4-pyranone and Gly116 and Gly117 of BChE.

### 3.5. Molecular Dynamics Simulation Assay

Molecular dynamics simulation is a computational approach that convincingly studies the flexibility of proteins and their roles in ligand binding [23]. In the process, interatomic forces are calculated via interaction potential, and system dynamics are understood by solving equations of motion. It samples conformational space and produces trajectories of molecular movements as a function of time. This technique is routinely applied to biological systems of pharmaceutical interest. All five compounds along with the standard control were examined in simulation studies and analyzed primarily via the RMSD statistical parameter (Figure 3) [24]. RMSD is the most acceptable and widely used measure of macromolecule structure and dynamics and provides the average distance between atoms of superimposed proteins. The RMSD values of the systems were in the following order: standard (0.66 Å) < **F5** (1.28 Å) < **F3** (1.31 Å) < **F1** (1.48 Å) < **F4** (1.49 Å) < **F2** (1.57 Å). The standard compound was revealed to show higher stability compared to the rest of the tested compounds. This demonstrated higher affinity of the standard molecule for AChE. Among the compounds, the **F5** conformation with AChE remained stable at the docked site. Simulation results clearly show analogies in the results with experimental data.

In the case of BChE complexes, very minor variations were noted in dynamics simulations with maximum RMSD around 4 Å (Figure 4). The receptor in the case of the **F3** compound was noted to have maximum flexibility though stable docked site conformation. These deviations were found due to flexible loop regions of the protein but could not affect the binding of the compound at the docked position. The **F2** complex showed better stability with an RMSD of 3.32 Å. The average RMSDs of the standard, **F4**, **F5**, and **F1** compounds were 3.8 Å, 3.7 Å, 3.6 Å, and 3.9 Å, respectively.

### 3.6. Estimation of Binding Free Energies

The MMGBSA and MMPBSA approaches are now commonly employed in the drug discovery process as they provide a cost-effective and easy-to-perform platform for estimating the real strength of interactions and stability of docked complexes [25]. Additionally, as docking predictions are not very reliable and require validation of a good binder to the receptor, a follow-up technique is strongly needed to complement the docking results. The simulation trajectories were scanned, and 100 snapshots were picked at regular intervals and analyzed. All of the compounds and control showed considerably negative net binding free energies, as listed in Table 2. This demonstrated the good binding affinity of the compounds for the receptors. The good net binding energy was the outcome of stable gas-phase energy in each complex, where both van der Waals and electrostatic forces contributed significantly.

### 3.7. Acute Toxicity

Results for **F1**–**F5** showed no mortality when animals were challenged with an oral dose of 600 mg/kg (b.w.). The animals in the groups were further tested at higher doses of 1000, 2000 and 3000 mg/kg (b.w.) and showed no physical or behavioral changes followed by effects on total weight and weight of the vital organs. Body weights and weights of vital organs (liver and kidney) did not show much difference when compared to the control group (Table 3).

The effects of flavonoids on biochemical parameters in acute toxicity assays (Table 4) were also assessed and found to be in normal ranges. Results from screening of the flavone derivatives for vital organ toxicity are given in Figure 5. Histological studies showed no abnormalities in the kidney and liver tissue in the flavonoid-treated groups when compared to the control. The preliminary pharmacological activity was also assessed at various dose concentrations of **F1**–**F5** (2.5–20 mg/kg b.w.) to determine the effective dose for assessment of cognitive function using the animal model. The preliminary pharmacological assessment of **F1**–**F5** showed promising results at a dose of 7.5 and 15 mg/kg in the behavioral model (Y-Maze) for memory (Appendix A). Hence, 7.5 and 15 mg/kg doses were selected for behavioral studies after preliminary pharmacological assessment.

### 3.8. Evaluation of Learning Behaviors

The behavioral task on the Y-Maze Paradigm and novel object discrimination (NOD) task were conducted to determine learning and memory functions.

### 3.9. Y-Maze Spontaneous Alternation

Table 5 show findings from Y-maze test; scopolamine significantly lessened the spontaneous alternation from 82.90% to 37.89% (*p* < 0.001, *n* = 8). The spontaneous alternation was normalized by donepezil, which increased the % alternation to 80.72 ± 1.59 (*p* < 0.001, *n* = 8). **F1**–**F5** at doses of 7.5 and 15 mg/kg also amplified the percent alternation. **F1** exhibited a considerable rise (*p* < 0.05, *p* < 0.01) in spontaneous alternation at 7.5 and 15 mg/kg as compared to the amnesic group, which was found to be 49.90 ± 1.33 (*p* < 0.05, *n* = 8) and 51.78 ± 1.45 (*p* < 0.01, *n* = 8). Among the flavonoids **F2**–**F5**, the most promising effects were produced by **F5** at 7.5 and 15 mg/kg as compared to the amnesic group, which was found to be 67.94 ± 1.51 (*p* < 0.001, *n* = 8) and 73.11 ± 1.63 (*p* < 0.001, *n* = 8), followed by **F4, F3** and **F2**. Moderate activity was observed in **F2** with 47.13 ± 1.65 (*p* < 0.05, *n* = 8) and 49.88 ± 1.28 (*p* < 0.05, *n* = 8) at doses of 7.5 and 15 mg/kg, respectively.

As shown in Table 5, administration of **F1**–**F5** (7.5 or 15 mg/kg) increased the % alternation in mice in a dose-dependent manner versus the amnesic group. These findings also confirmed that there were no significant differences in % alternation after administering **F1**–**F5** (7.5 or 15 mg/kg) or donepezil (2 mg/kg). The above results suggest that **F5** is more potent in comparison to other flavonoids.

### 3.10. Novel Object Discrimination Task

Results of memory enhancing potentials for long-term memory over NOD task model are given in Figure 6. In the sample phase for all tested groups, no changes were observed in exploration time (s) for the objects. In the (T1) test phase, the exploration time in sec was considerably greater for novel object (NO) than similar (identical) object in groups that were treated with **F1**–**F5** at 7.5 and 15 mg/kg and donepezil standard (2 mg/kg).

The discrimination index (%DI) in the novel object discrimination task (NODT) was significantly lower in the scopolamine-induced amnesic mice, which was found to be 30.27 ± 1.61 (*p* < 0.001, *n* = 8) versus the normal control group (72.80 ± 1.51). Administration of synthetic flavonoids (**F1**–**F5**) at 7.5 and 15 mg/kg b.w. significantly prevented this reduction and enhanced (*p* < 0.05, *p* < 0.01, *p* < 0.001) the index when compared to the amnesic group, as shown in Figure 6. The administration of 7.5 mg/kg b.w. of **F1** exhibited a 48.67 ± 1.57 %DI, and 15 mg/kg b.w. produced a 51.11 ± 1.48 %DI response. The **F5** at 7.5 mg/kg b.w. produced a significant %DI of 59.76 ± 1.24 (*p* < 0.001, *n* = 8) versus the amnesic group (30.27 ± 1.61), while **F5** displayed promising results (*p* < 0.001) with a %DI of 62.12 ± 1.28 at a dose of 15 mg/kg b.w. versus the amnesic group. The administration of **F2**–**F4** at 7.5 and 15 mg/kg b.w. also produced a significant %DI versus the amnesic group, but to a lesser extent in comparison to **F5**. The %DI in the novel object discrimination task (NODT) was significantly higher for standard donepezil, which was found to be 71.39 ± 1.59%, *p* < 0.001, *n* = 8 versus the amnesic control group.

As shown in Figure 6**,** administration of **F1**–**F5** (7.5 or 15 mg/kg) increased the discrimination index (%DI) in mice in a dose-dependent manner versus the amnesic group. These findings also confirmed that there were no significant differences among the discrimination indexes (%DI) after administering **F1**–**F5** (7.5 or 15 mg/kg) or donepezil (2 mg/kg).

### 3.11. Assessment of Antioxidant Enzyme Activities on Oxidative Stress

Administration of scopolamine caused a substantial elevation of the AChE level, decreased the level of ACh, and augmented oxidative stress in mice, as evidenced from decreases in the CAT and SOD levels in the brain. Compounds **F1**–**F5** showed a distinct effect on these alterations by decreasing the level of AChE. Likewise, they also enhanced the levels of ACh, CAT and SOD, signifying their possible roles as antioxidants in oxidative stress.

The results in Table 6 show the significant output of **F1**–**F5** on catalase (CAT) level in the brains of studied animals. In contrast, with the control level of catalase, 32.21 ± 1.37, scopolamine administration caused a significant decrease, to 7.08 ± 1.13, *p* < 0.001, in the level of catalase enzyme. **F1**–**F5** produced a similar response to standard and significantly increased the level of catalase at doses of 7.5 and 15 mg/kg b.w. in comparison to the amnesic group. **F1** significantly increased the level of catalase (21.93 ± 1.37 and 22.31 ± 1.22) at doses of 7.5 and 15 mg/kg b.w., respectively (Table 6). **F5** produced a similar response to standard and significantly increased the level of catalase (24.18 ± 1.41 and 24.91 ± 1.29) at doses of 7.5 and 15 mg/kg b.w. when compared to the amnesic group. Other synthetic flavonoids (**F2**–**F4**) significantly increased the level of catalase at a dose of 7.5 mg/kg b.w. Donepezil increased the level significantly, to 30.88 ± 1.44, *p* < 0.001, *n* = 8.

Scopolamine administration resulted in significant decline in the level of SOD, to 5.19 ± 0.91 units/mg of protein, *p* < 0.001, *n* = 8 in the brain homogenate compared to control (14.11 ± 1.08 units/mg protein, *n* = 8) (Table 6). This decline was overturned by the mice pre-treated with standard donepezil and was documented at 12.95 ± 1.47 unit/mg protein, *p* < 0.001, *n* = 8 in the brain homogenate when compared to the amnesic group.

Pretreatment of mice with **F1**–**F5** significantly increased the SOD level in the brain. Among synthetic flavonoids, **F5** was found to be the most promising and increased the SOD level significantly in the brain to 9.88 ± 1.49 and 10.20 ± 1.62 units/mg protein, respectively, at the doses of 7.5 and 15 mg/kg b.w. *p* < 0.001, *n* = 8 in comparison with the scopolamine-treated group. **F1** showed less response in comparison to other flavonoids, and the SOD level was found to be 8.97 ± 1.12 and 9.12 ± 1.28 units/mg protein, respectively, at the doses of 7.5 and 15 mg/kg b.w. Similarly, the levels of MDA and GSH were overturned significantly by **F1**–**F5** and standard donepezil, as documented in Table 6.

As shown in Table 6**,** administration of **F1**–**F5** (7.5 or 15 mg/kg) produced maximum declines in the MDA level and a considerable rise in the level of SOD, CAT and GSH in mice brains in a dose-dependent manner versus the amnesic group. These findings also confirmed that there were no significant differences among the biomarker levels in the brain after administering **F1**–**F5** (7.5 or 15 mg/kg) or donepezil (2 mg/kg).

### 3.12. Effect on AChE and ACh Levels

A considerable rise was observed in the level of AChE (33.11 ± 0.77, ^###^ *p* < 0.001, *n* = 8) in the brain homogenate after scopolamine administration in comparison to the control group with 13.38 ± 0.59, *n* = 8 (Figure 7); this was effectively reversed by donepezil and **F1**–**F5,** signifying their function in the treatment of memory impairment probably via ChE inhibition. The maximum declines in AChE level were produced by administration of **F5** at doses of 7.5 and 15 mg, and were found to be 15.34 ± 0.69, *** *p* < 0.001, *n* = 8 and 14.67 ± 0.67, *** *p* < 0.001, *n* = 8, respectively, when compared to the amnesic (scopolamine) group. Among the flavones, **F1** was found to be less effective and produces a decline of 21.31 ± 0.53, ** *p* < 0.001, *n* = 8 and 21.01 ± 0.61, ** *p* < 0.001, *n* = 8 at the doses of 7.5 and 15 mg/kg b.w., respectively, when compared to the amnesic (scopolamine) group. Donepezil produced a significant fall in the level of AChE at a dose of 2 mg/kg b.w., to 12.77 ± 0.58, *** *p* < 0.001, *n* = 8 when compared to the amnesic (scopolamine) group. Simultaneously, a significant descent in the ACh level was also noted in the amnesic group, which was reversed by analyzing the data of the groups treated with samples and standard.

As shown in Figure 7**,** administration of **F1**–**F5** (7.5 or 15 mg/kg) produced maximum decreases in the AChE level and a considerable increase in the level of ACh in mice brain in a dose-dependent manner versus the amnesic group. These findings also confirmed that there were no significant differences in the levels of AChE and ACh after administering **F1**–**F5** (7.5 or 15 mg/kg) or donepezil (2 mg/kg).

## 4. Discussion

Flavonoids are natural secondary metabolites abundantly present in vegetables and fruits, and have recently attracted greater interest in the scientific community. They have shown a large number of pharmacological activities in neurological disorders such as neuro-protective effects [26], inhibition of Ab fibril formation [27], AChE inhibition [28], free radical scavenging [29], and metal-chelating potential [30].

The mechanistic and molecular level studies of AD have suggested its association with many pathological conditions, of which the most important are amyloid-beta (Ab) plaques, neurofibrillary tangles encompassing hyperphosphorylated and aggregated tau protein, neuroinflammation and neurodegenerative disease [31,32,33,34,35,36]. In clinical trials, it was noted that the acetylcholine level can be retained if the two acetylcholinesterases AChE and BuChE are inhibited. Medications such as NMDAR antagonists and acetylcholinesterase (AChE) inhibitors have been approved by the US FDA for the management of Alzheimer’s disease [37,38]. The currently used treatment for AD is focused only on relieving the symptoms without targeting the responsible factors of the disease. Therefore, an attempt has been made here to design and synthesize flavonoid derivatives with the potential to inhibit the target enzymes, AChE and BuChE, which would probably be an excellent strategy in devising an anti-Alzheimer drug. Out of the compounds, **F5,** demonstrated excellent activity against AChE as compared to other tested members of the prepared series (Table 1), whereas **F2** was more potent against BuChE followed by **F5** with a slight difference. The observed potential of the compounds against AChE and BuChE was validated through a molecular docking approach where **F5** (Figure 1a), the most active compound, exhibited the most favorable interactions with the enzyme molecule involving hydrophobic interaction between 4-pyranone and Trp86 of AChE and a halogen bond between chlorophenyl and Ser203. The additional non-covalent halogen bond interaction was present in compound **F5,** validating its high experimental activity. It has been previously reported that the flavonoid scaffold could interact with the PAS of AChE via aromatic stacking interactions [39]. In the case of BuChE, the docking interactions were favorable with compound **F2** (Figure 2a), where pi-cation interaction between 4-pyranone and water molecules along with the establishment of two halogen bonds between the chlorophenyl and water molecules were observable, thus validating the in vitro experimental results. Among the flavonoids **F1**–**F5**, the most promising effect was produced by **F5** at 7.5 and 15 mg/kg and increased the spontaneous alternation in the Y-maze to 67.94 ± 1.51 (*p* < 0.001, *n* = 8) and 73.11 ± 1.63 (*p* < 0.001, *n* = 8) as compared to the amnesic group (37.89 ± 1.61, *p* < 0.001, *n* = 8). **F5** at 7.5 mg/kg b.w. produced a significant discrimination index (%DI) of 59.76 ± 1.24 (*p* < 0.001, *n* = 8) in the novel object discrimination task (NODT) versus the amnesic group (30.27 ± 1.61), while **F5** displayed promising results (*p* < 0.001) with a %DI of 62.12 ± 1.28 at a dose of 15 mg/kg b.w. versus the amnesic group. The results suggest that **F5** is more potent in comparison to other flavonoids (Table 5 and Figure 6).

Tacrine (Cognex™, Natick, MA, USA), an AChE inhibitor, was the first drug approved by the Food and Drug Administration, in 1993, for the treatment of AD, but due to its hepato-toxic potential, the drug was withdrawn from the market in 2012 [40]. Administration of scopolamine causes substantial elevation of the AChE level, decreases the ACh level, and augments the oxidative stress in experimental mice as evidenced by decreased CAT and SOD levels in the brain. **F1**–**F5** showed a distinct effect on these alterations by reducing the content of AChE, MDA and GSH. It also elevated the levels of ACh, CAT and SOD, signifying their possible roles as antioxidants in relieving oxidative stress. The maximum declines in the AChE level were produced by administration of **F5** at doses of 7.5 and 15 mg, and were found to be 15.34 ± 0.69, *** *p* < 0.001, *n* = 8 and 14.67 ± 0.67, *** *p* < 0.001, *n* = 8, respectively, when compared to the amnesic (scopolamine) group (Figure 7). AChE inhibitors such as donepezil, rivastigmine, and galantamine are among commonly prescribed medications that were approved in 1996, 2000, and 2001, respectively, for the treatment of AD [32,33]. The only AChE inhibitor that is approved to treat all stages of Alzheimer’s disease is donepezil [32,33,40]. Of these three drugs, two are plant secondary metabolites, which is why we tested the flavonoid derivatives in this study.

To further validate the observed anticholinesterase potential, ex vivo studies on experimental animal brain were performed. The results for the biomarkers in Table 6 show profound effects on the catalase and superoxide dismutase levels in the brains of the studied animals. **F5** produced a response similar to that of the used standard and significantly increased the levels of catalase (24.18 ± 1.41 and 24.91 ± 1.29) and SOD (9.88 ± 1.49 and 10.20 ± 1.62 units/mg protein at doses of 7.5 and 15 mg/kg b.w., respectively) as compared to the levels in the amnesic group animals. Similarly, the levels of MDA and GSH were lowered significantly by **F1**–**F5** and standard donepezil (Table 6). These findings also confirmed that there were no significant differences among the in vivo and ex vivo results after administration of **F1**–**F5** (7.5 or 15 mg/kg) or donepezil (2 mg/kg).

The second strategy for treating AD is the use of an NMDAR antagonist that has also been approved by the FDA. There is an excessive release of glutamate from damaged cells in AD, which causes calcium influx to neurons through activation of NMDAR, resulting in excitotoxicity that finally leads to neuronal death. The NMDAR antagonist memantine (Namenda™), approved in 2003, acts as a neuroprotective medication for the treatment of moderate to severe conditions [32,33]. Memantine can be only prescribed or given in combination with AChE inhibitors such as donepezil [40].

As is clear from the above discussion, acetylcholine inhibitors are needed even if a second strategy is used. So far, 100% efficient inhibitors of acetylcholinesterases have not been identified, and research is in progress to find an efficient inhibitor of these enzymes. The present study was an attempt in this connection, and its results indicate that these flavonoid derivatives are capable of inhibiting these enzymes, although there were significant differences in in vitro activities of the standard and tested compounds (which need to be improved by further derivatization). Nevertheless, the in vivo and ex vivo results were satisfactory, indicating that these compounds may be considered as potent neuro-pharmacological drug candidates.

## 5. Conclusions

In the current study, the compounds abbreviated as **F1**–**F5** were evaluated for their anticholinesterase potential; out of all tested compounds, **F5** showed the most potent activity against AChE. The observed potential of **F5** was further supported by molecular docking study. The in vitro and computational approaches suggested that **F5** could serve as a potent inhibitor of cholinesterases and might serve as a lead molecule for structural optimization to further enhance its activity. As the most potent compound, **F5** was further subjected to in vivo anti-amnesic evaluation in animal models. **F5** showed significant anti-amnesic effects in the scopolamine-induced amnesic model. The administration of the flavonoid ameliorated memory loss in behavioral models, along with appreciable protection from oxidative stress in the brains of scopolamine-induced amnesic mice. The results from this pre-clinical study indicate that the synthetic flavonoids may be useful in the development of new and potent neuro-pharmacological drug candidates.

## Figures and Tables

**Figure 1 brainsci-12-00731-f001:**
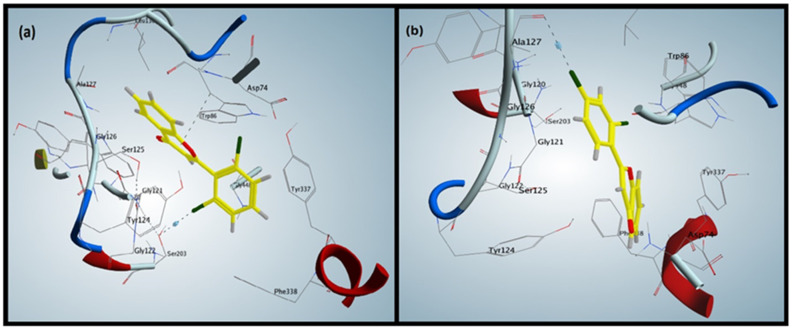
(**a**) shows the docking interactions of the most active compound, **F5.** (**b**) shows the interaction of the least active compound, **F2**.

**Figure 2 brainsci-12-00731-f002:**
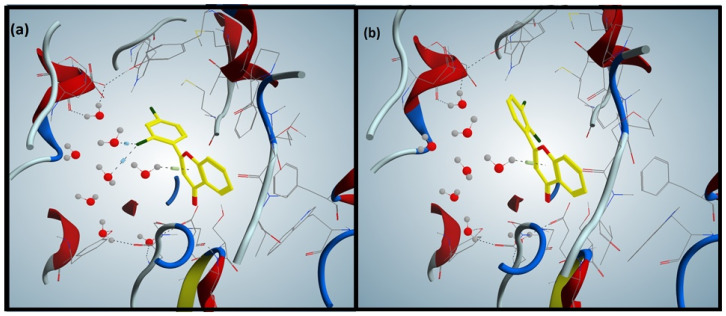
(**a**) shows the docking interactions of the most active compound, **F2.** (**b**) shows the interaction of the least active compound, **F3**.

**Figure 3 brainsci-12-00731-f003:**
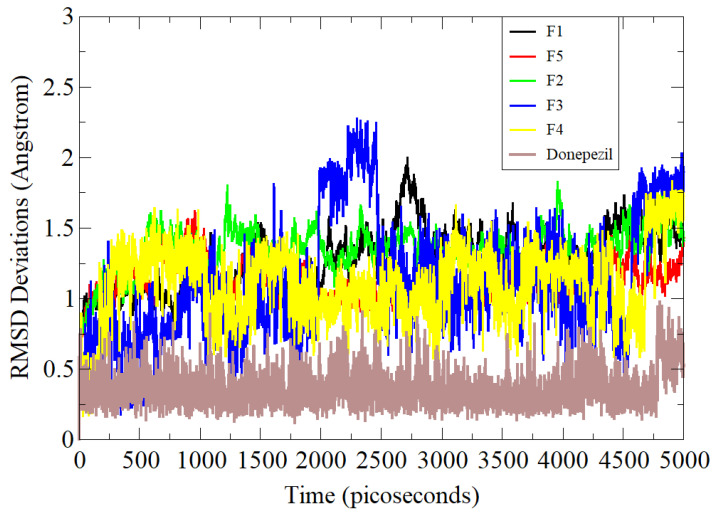
Simulation-based RMSD analysis of tested compounds and standard control AChE complexes.

**Figure 4 brainsci-12-00731-f004:**
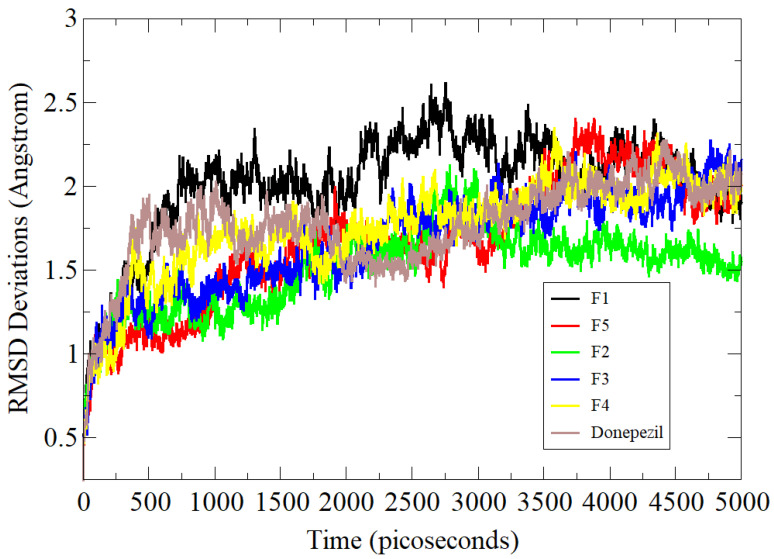
Simulation-based RMSD analysis of tested compounds and standard control BChE complexes.

**Figure 5 brainsci-12-00731-f005:**
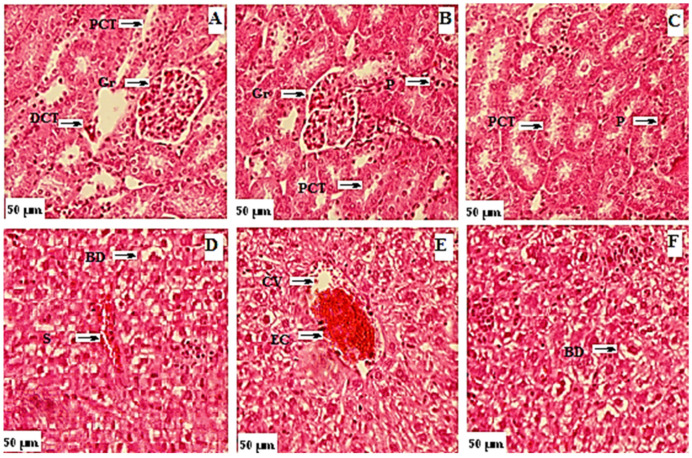
Photomicrograph of kidney and liver from toxicity study of flavonoids. Kidney: (**A**) control group; (**B**) group treated with 3000 mg/kg b.w. of **F1**; (**C**) group treated with 3000 mg/kg b.w. of **F5. Gr:** glomeruli, **PCT:** proximal convoluted tubule, **DCT**: distal convoluted tubule, **P:** podocyte. Liver: (**D**) control group; (**E**) group treated with 3000 mg/kg b.w. of **F1**; (**F**) group treated with 3000 mg/kg b.w. of **F5. BD:** bile duct, **S:** sinusoids, **EC**: endothelial cells, **CV**: central vein.

**Figure 6 brainsci-12-00731-f006:**
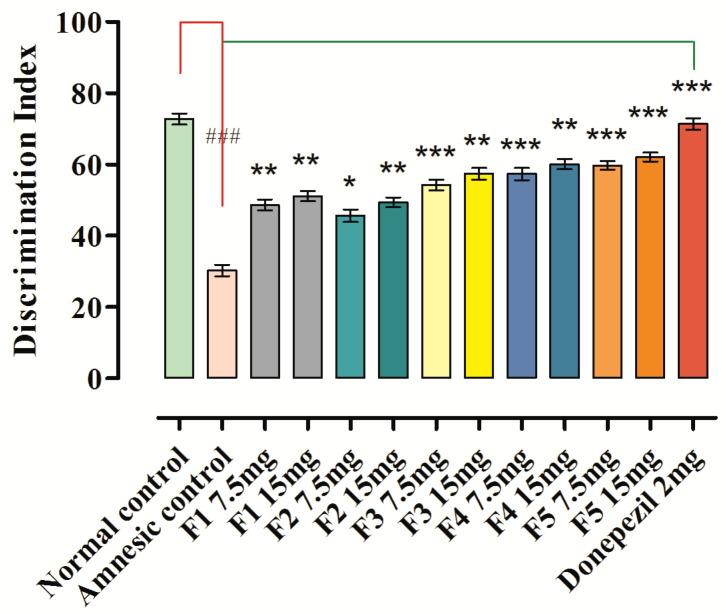
Behavioral results from NODT in memory for **F1**–**F5** with index of % Discrimination. Mean ± SEM (*n* = 8). One-way ANOVA followed by Dunnett’s post hoc multiple comparison test to determine the values of *p*. ^###^ *p* < 0.001 comparison of amnesic (scopolamine) group vs. normal control. * *p* < 0.05, ** *p* < 0.01 and *** *p* < 0.001 as comparison of **F1**–**F5** treated groups, donepezil treated group vs. amnesic group (scopolamine) using one-way ANOVA followed by Dunnett’s comparison.

**Figure 7 brainsci-12-00731-f007:**
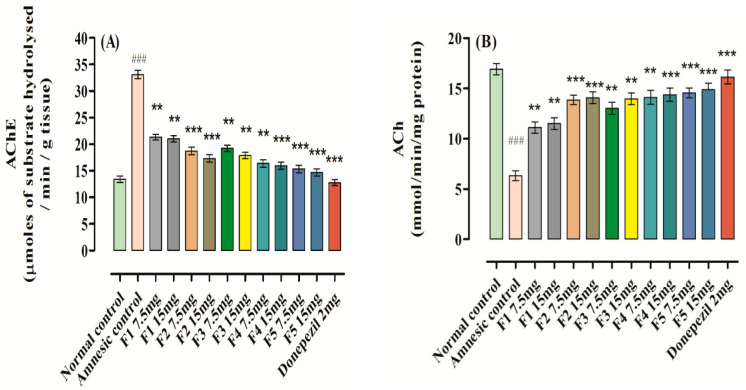
Effect of **F1**–**F5** on (**A**) AChE activity and (**B**) ACh level. Mean ± SEM (*n* = 8). Statistically analyzed with one-way ANOVA followed by Dunnett’s post hoc multiple comparison test to determine the values of *p*. ^###^ *p* < 0.001 comparison of amnesic (scopolamine) group vs. normal control. ** *p* < 0.01 and *** *p* < 0.001 as comparison of **F1**–**F5** treated groups, donepezil-treated group vs. amnesic group (scopolamine) using one-way ANOVA followed by Dunnett’s comparison.

**Table 1 brainsci-12-00731-t001:** Synthesized flavones with IC_50_ values against cholinesterase enzymes.

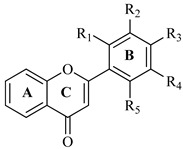						
**Flavone**	**R_1_**	**R_2_**	**R_3_**	**R_4_**	**IC_50_ (** **μ** **g/mL)**	
					**AChE**	**BChE**
**F1**	H	H	Cl	H	165.21 ± 1.53	154.71 ± 1.38
**F2**	Cl	H	Cl	H	131.33 ± 1.12	105.20 ± 1.43
**F3**	Cl	Cl	H	H	126.29 ± 1.33	147.12 ± 1.23
**F4**	H	Cl	Cl	H	112.33 ± 1.16	121.77 ± 1.19
**F5**	Cl	H	H	Cl	98.42 ± 0.97	109.61 ± 1.11
Donepezil					4.91 ± 0.51	3.98 ± 0.67

**Table 2 brainsci-12-00731-t002:** Cholinesterase (AChE and BuChE) inhibitory activities of flavones.

Sample Test	AChE IC_50_ (µg/mL)	Net Binding Energy (Kcal/mol)		BChE IC_50_ (µg/mL)	Net Binding Energy (Kcal/mol)	
		MMGBSA	MMPBSA		MMGBSA	MMPBSA
**F1**	165.21 ± 1.53	−42.31	−41.15	154.71 ± 1.38	−35.64	−31.45
**F2**	131.33 ± 1.12	−45.32	−43.87	105.20 ± 1.43	−49.74	−51.42
**F3**	126.29 ± 1.33	−46.87	−50.78	147.12 ± 1.23	−43.48	−38.61
**F4**	112.33 ± 1.16	−50.99	−48.65	121.77 ± 1.19	−48.19	−43.64
**F5**	98.42 ± 0.97	−49.45	−53.14	109.61 ± 1.11	−51.88	−52.78
**Donepezil**	4.91 ± 0.51	−55.91	−54.51	3.98 ± 0.67	−40.17	−43.14

**Table 3 brainsci-12-00731-t003:** Effects on total weight and weight of vital organs in mice measured by acute toxicity study.

Treatment/Dose (mg)		Total Weight (g)	Kidney Weight (g)	Liver Weight (g)
**Normal Control**		22.68 ± 1.40	0.31 ± 0.13	1.22 ± 0.27
**F1**	600	21.82 ± 1.39	0.29 ± 0.19	1.24 ± 0.21
	1000	22.93 ± 1.91	0.29 ± 0.18	1.23 ± 0.26
	2000	21.79 ± 1.72	0.30 ± 0.20	1.23 ± 0.24
	3000	22.71 ± 1.79	0.31 ± 0.27	1.25 ± 0.29
**F5**	600	20.97 ± 1.66	0.30 ± 0.21	1.24 ± 0.25
	1000	21.88 ± 1.89	0.31 ± 0.24	1.22 ± 0.29
	2000	22.91 ± 1.67	0.29 ± 0.20	1.26 ± 0.21
	3000	22.65 ± 1.91	0.30 ± 0.19	1.23 ± 0.25

Mean ± SEM (*n* = 3).

**Table 4 brainsci-12-00731-t004:** The effect of flavonoids on biochemical parameters in acute toxicity assay.

Treatment/Dose (mg)		Glucose (mg/dL)	Triglyceride (mg/dL)	AST (U/L)	ALP (U/L)	Creatinine (mg/dL)	Total Cholesterol (mg/dL)
**Normal control**		84.72 ± 2.41	94.39 ± 2.13	107.62 ± 1.87	132.98 ± 3.47	0.25 ± 0.04	99.67 ± 1.79
**F1**	600	85.51 ± 2.33	95.20 ± 2.76	107.89 ± 2.01	132.54 ± 2.78	0.36 ± 0.05	103.19 ± 2.06
	1000	86.98 ± 2.09	94.78 ± 1.98	108.34 ± 2.66	136.33 ± 2.39	0.38 ± 0.06	102.76 ± 1.87
	2000	86.45 ± 2.27	96.22 ± 1.70	109.73 ± 1.98	135.29 ± 4.17	0.34 ± 0.04	102.05 ± 2.01
	3000	85.72 ± 1.97	97.41 ± 1.83	109.90 ± 2.09	136.09 ± 3.12	0.29 ± 0.03	101.28 ± 1.98
**F5**	600	86.98 ± 2.12	95.31 ± 2.61	108.37 ± 1.97	134.41 ± 2.81	0.28 ± 0.04	102.91 ± 1.93
	1000	87.01 ± 2.21	96.08 ± 2.01	107.41 ± 2.03	135.87 ± 2.24	0.35 ± 0.05	103.02 ± 1.79
	2000	86.89 ± 1.98	95.37 ± 1.98	108.39 ± 1.89	133.91 ± 3.71	0.33 ± 0.06	102.73 ± 2.10
	3000	85.34 ± 2.01	96.50 ± 2.05	107.76 ± 1.92	134.02 ± 3.28	0.30 ± 0.04	103.09 ± 1.83

Mean ± SEM (*n* = 3).

**Table 5 brainsci-12-00731-t005:** Effect on % Spontaneous alternation performance of **F1**–**F5** in behavioral Y-maze test.

Treatment/Dose (mg)		Spontaneous Alternation Performance
**Normal control**		82.90 ± 1.67
**Amnesic control (Scopolamine)**		37.89 ± 1.61 ^###^
**F1**	7.5	49.90 ± 1.33 *
	15	51.78 ± 1.45 **
**F2**	7.5	47.13 ± 1.65 *
	15	49.88 ± 1.28 *
**F3**	7.5	59.61 ± 1.61 **
	15	65.78 ± 1.57 ***
**F4**	7.5	57.92 ± 1.41 **
	15	66.70 ± 1.63 ***
**F5**	7.5	67.94 ± 1.51 ***
	15	73.11 ± 1.63 ***
**Donepezil**	2	80.72 ± 1.59 ***

Mean ± SEM (*n* = 8). Statistically analyzed with one-way ANOVA followed by Dunnett’s post hoc multiple comparison test to determine the values of *p*. ^###^ *p* < 0.001 comparison of amnesic (scopolamine) group vs. normal control group. * *p* < 0.05, ** *p* < 0.01 and *** *p* < 0.001 as comparison of **F1**–**F5** treated groups and donepezil-treated group vs. amnesic group (scopolamine) using one-way ANOVA followed by Dunnett’s comparison.

**Table 6 brainsci-12-00731-t006:** Effects on biomarker levels in the brain.

Sample Test (mg)		SOD (U/mg of Protein)	CAT (U/mg of Protein)	MDA (nmol/mg Protein)	GSH (μg/mg of Protein)
**Control**		14.11 ± 1.08	32.21 ± 1.37	8.99 ± 1.01	45.87 ± 1.41
Amnesic control		5.18 ± 0.91 ^###^	7.08 ± 1.13 ^###^	26.60 ± 1.41 ^###^	15.06 ± 1.37 ^###^
**F1**	7.5	8.97 ± 1.12 **	21.93 ± 1.37 **	16.39 ± 1.28 **	34.62 ± 1.50 **
	15	9.12 ± 1.28 **	22.31 ± 1.22 **	16.12 ± 1.47 ***	35.70 ± 1.47 **
**F2**	7.5	9.25 ± 1.36 **	22.83 ± 1.19 ***	15.16 ± 1.21 **	34.93 ± 1.22 **
	15	9.33 ± 1.45 **	23.71 ± 1.45 ***	14.90 ± 1.33 ***	35.91 ± 1.39 ***
**F3**	7.5	9.19 ± 1.21 **	22.97 ± 1.71 **	16.37 ± 1.13 **	35.61 ± 1.55 **
	15	9.41 ± 1.29 ***	24.50 ± 1.41 ***	15.94 ± 1.65 **	36.98 ± 1.66 ***
**F4**	7.5	9.30 ± 1.41 ***	22.21 ± 1.40 **	15.74 ± 1.21 **	36.21 ± 1.35 ***
	15	9.71 ± 1.56 ***	23.39 ± 1.32 **	15.01 ± 1.37 **	38.30 ± 1.40 ***
**F5**	7.5	9.88 ± 1.49 ***	24.18 ± 1.41 ***	14.98 ± 1.22 ***	37.28 ± 1.60 ***
	15	10.20 ± 1.62 ***	24.91 ± 1.29 ***	14.48 ± 1.38 **	39.01 ± 1.39 ***
Donepezil	2	12.95 ± 1.47 ***	30.88 ± 1.44 ***	10.95 ± 1.28 ***	45.09 ± 1.30 **

Mean ± SEM (*n* = 8). Statistically analyzed with one-way ANOVA followed by Dunnett’s post hoc multiple comparison test to determine the values of *p*. ^###^ *p* < 0.001 comparison of amnesic (scopolamine) group vs. normal control. ** *p* < 0.01 and *** *p* < 0.001 as comparison of **F1**–**F5** treated groups, donepezil-treated group vs. amnesic group (scopolamine) using one-way ANOVA followed by Dunnett’s comparison.

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
