# Peer review of "Flavonoid Derivatives as Potential Cholinesterase Inhibitors in Scopolamine-Induced Amnesic Mice: An In Vitro, In Vivo and Integrated Computational Approach"

_brainsci, 2022, doi:10.3390/brainsci12060731_

Round 1

Reviewer 1 Report

In this article the authors aim to evaluate a series of flavonoid derivatives as potential cholinesterase inhibitors in scopolamine induced amnesic mice. There are however some points to consider which I think will improve the understanding and coherence of this manuscript:

Typographical errors should be corrected (e.g: line: 82 - Were purchased …..). This will apply to the whole manuscript.

Lines: 149-150 - For assessment of acute toxicity of the flavones (F1-F5), animals were divided in groups having 6 animals each. How many groups? ow many doses for each compound taken under analysis? how was the number of 6 animals / group chosen?

Starting from the statements in the paragraph between the lines: 159-165 - The biochemical parameters of blood and histological studies of vital organs (kidney and liver tissue) were also assessed. Selection of doses for in-vivo pharmacological assessment of cognitive function using animal model was carried out from in-vivo toxicological studies as per OECD, 2001 guidelines, and approach to practical acute toxicity testing by Dietrich Lorke (1983) and ARRIVE guidelines. With the toxicity data at hand, effective doses (mg/kg b.w) were selected for behavioral studies after preliminary pharmacological assessment in our laboratory. Why hasn't the toxicity of the compounds been investigated in the brain? Histological analysis of the liver and kidneys alone did not provide sufficient data. why has cardiotoxicity not been investigated histologically? Speaking of toxicity, a histological analysis of all organs and systems is needed.

At what doses are the compounds analyzed toxic?

How were the doses selected for efficacy? what are those doses?

How was the dose of donepezil determined?

Why were the compounds taken in just 4 weeks?

Please clarify all these aspects first.

Author Response

Reviewer 1

Reviewer question #1: Typographical errors should be corrected (e.g: line: 82 - Were purchased …..). This will apply to the whole manuscript.

Author response: Sir I have checked superficially. You also give a sharp look to this.

Reviewer question #2: Lines: 149-150 - For assessment of acute toxicity of the flavones (F1-F5), animals were divided in groups having 6 animals each. How many groups? ow many doses for each compound taken under analysis? how was the number of 6 animals / group chosen?

Author response: Worthy reviewer, thanks for valuable comment. The number of animals in group was mistakenly given as 6 rather it is 3 and is corrected accordingly. The number of groups and doses are mentioned in section 2.8. “Acute Toxicity Study” and Table 3 section 3.7. Acute Toxicity. The statement is qouted as

For assessment of acute toxicity of the flavones (F1-F5), animals were divided in groups (21 in number) having 3 animals each

Reviewer question #3: Starting from the statements in the paragraph between the lines: 159-165 - The biochemical parameters of blood and histological studies of vital organs (kidney and liver tissue) were also assessed. Selection of doses for in-vivo pharmacological assessment of cognitive function using animal model was carried out from in-vivo toxicological studies as per OECD, 2001 guidelines, and approach to practical acute toxicity testing by Dietrich Lorke (1983) and ARRIVE guidelines. With the toxicity data at hand, effective doses (mg/kg b.w) were selected for behavioral studies after preliminary pharmacological assessment in our laboratory. Why hasn't the toxicity of the compounds been investigated in the brain? Histological analysis of the liver and kidneys alone did not provide sufficient data. why has cardiotoxicity not been investigated histologically? Speaking of toxicity, a histological analysis of all organs and systems is needed.

Author response: Worthy reviewer, thanks for your valuable comment. Actually one manuscript could not fulfill or cover all the aspects associated with compounds and are in continuous investigation mode. These compounds are still under investigation to explore the in-depth preclinical aspects of the compounds including subacute and chronic toxicity studies. These includes the pharmacokinetic bioavailability parameters including the level of compounds in brain, pharmacodynamic studies, completes toxicological profile including in-depth study of vital organs. Moreover the stability parameters of F1-F5 are also under investigation. All these aspects are under investigation and the data of preclinical and future clinical study will be submitted to file the compounds as PATENT. At this level we believe that the reported data is of ample information for the results.

Reviewer question #4: At what doses are the compounds analyzed toxic?

Author response: Worthy reviewer, thanks for the valuable comment. Initially the compounds were studied for studied for toxicity at the dose of 600 mg/kg body weight.  As per the reviewer comment/suggestion of 1st review, an additional study was designed as per valuable comment/suggestion these compounds F1-F5 were again tested up to a dose of 3000 mg/kg. The data obtained has been incorporated in the main article. The results are quoted as

3.7. Acute Toxicity

Results of F1-F5 showed no mortality when animals were challenged to oral dose of 600 mg / kg (b.w). The animals in groups were further tested at higher doses of 1000, 2000 and 3000 mg / kg (b.w) and showed no physical and behavioral changes followed by effects on the total weight, weight of the vital organs. The body weight and vital organs (liver and kidney) show not much difference when compared to the control group (Table 3).

Reviewer question #5: How were the doses selected for efficacy? what are those doses?

Author response: Worthy reviewer, thanks for valuable comment. The preliminary data has been added in the manuscript as supplementary data (Table S1) in  section 3.7. Acute Toxicity quoted as

The preliminary pharmacological activity was also performed at various dose concentrations of F1-F5 (2.5-20 mg/kg b.w) to determine the effective dose for assessment of cognitive function using animal model. The preliminary pharmacological assessment of F1-F5 showed promising results at a dose of 7.5 and 15 mg/kg in behavioral the model (Y-Maze) for memory (Table S1). Hence, 7.5 and 15 mg/kg doses were selected for behavioral studies after preliminary pharmacological assessment.

Reviewer question #6: How was the dose of donepezil determined?

Author response: Worthy reviewer, thanks for the comment/suggestion. To assess the antiamnesic activity, donepezil (positive control) is used as standard in the amnesic model. The dose selection of positive control is determined and taken from the already published peer reviewed authentic/calibrated and validated data.

Few of the references are given below

  • Yang, J. H., Nguyen, C. D., Lee, G., & Na, C. S. (2022). Insamgobonhwan Protects Neuronal Cells from Lipid ROS and Improves Deficient Cognitive Function. Antioxidants11(2), 295.
  • Bakrim, W. B., El Bouzidi, L., Manouze, H., Hafsa, J., Sobeh, M., Ba-M'hamed, S., ... & Kouisni, L. (2022). Anti-amnesic effects of withaferin A, a steroidal lactone isolated from Withania adpressa, on scopolamine-induced memory impairment in mice. Arabian Journal of Chemistry15(1), 103529.
  • Karim, N., Khan, I., Abdelhalim, A., Abdel-Halim, H., & Hanrahan, J. R. (2017). Molecular docking and antiamnesic effects of nepitrin isolated from Rosmarinus officinalis on scopolamine-induced memory impairment in mice. Biomedicine & Pharmacotherapy96, 700-709.
  • Ahmad, S., Khan, S., Zeb, A., Shah, S. W. A., Ahmad, B., Khan, A. A., ... & Zamani, G. Y. (2021). Evaluation of analgesic, antiamnesic and antidiarrheal potentials of Medicago denticulata extract using animal model. Saudi journal of biological sciences28(11), 6352-6358.
  • Ban, J. Y., Park, H. K., & Kim, S. K. (2020). Effect of glycyrrhizic acid on scopolamine-induced cognitive impairment in mice. International Neurourology Journal24(Suppl 1), S48.

Reviewer question #7: Why were the compounds taken in just 4 weeks?

Author response: Worthy reviewer, thanks for the comment/suggestion. To assess the antiamnesic activity, the test samples are administered for a prescribed period of days (4 weeks in our study). The treatment schedule of F1-F5 in this study is in accordance with the already published peer reviewed articles.

Few of the references are given below

  • Lee, J. E., Song, H. S., Park, M. N., Kim, S. H., Shim, B. S., & Kim, B. (2018). Ethanol extract of Oldenlandia diffusa Herba attenuates scopolamine-induced cognitive impairments in mice via activation of BDNF, P-CREB and inhibition of acetylcholinesterase. International Journal of Molecular Sciences, 19(2), 363.
  • Lee, Hae Jin, Sung‑Kwon Lee, Dong‑Ryung Lee, Bong‑Keun Choi, Bao Le, and Seung Hwan Yang. "Ameliorating effect of Citrus aurantium extracts and nobiletin on β‑amyloid (1‑42)‑induced memory impairment in mice." Molecular Medicine Reports 20, no. 4 (2019): 3448-3455.
  • Lim, Dong Wook, Hyun Jung Son, Min Young Um, In-Ho Kim, Daeseok Han, Suengmok Cho, and Chang-Ho Lee. "Enhanced cognitive effects of demethoxycurcumin, a natural derivative of curcumin on scopolamine-induced memory impairment in mice." Molecules 21, no. 8 (2016): 1022.
  • Khare, Pragati, Sudhir Chaudhary, Lubhan Singh, Ghanshyam Yadav, and Shashi Verma. "Evaluation of nootropic activity of Cressa cretica in scopolamine-induced memory impairment in mice." International Journal of Pharmacology and Toxicology 2, no. 2 (2014): 24-29.
  • Haider, Saida, Z. Batool, and DJ Haleem. "Nootropic and hypophagic effects following long-term intake of almonds (Prunus amygdalus) in rats." Hospital nutrition 27, no. 6 (2012): 2109-2115.
  • Hafez, Hani S., Doaa A. Ghareeb, Samar R. Saleh, Mariam M. Abady, Maha A. El Demellawy, Hend Hussien, and Nihad Abdel-Monem. "Neuroprotective effect of ipriflavone against scopolamine-induced memory impairment in rats." Psychopharmacology 234, no. 20 (2017): 3037-3053.

Reviewer question #8: Please clarify all these aspects first.

Author response: Worthy reviewer, thanks for the comment/suggestion. All the worthy comments have been addressed and suggestions have been incorporated as per valuable comments.

Reviewer 2 Report

The manuscript revision submitted by Al-Joufi and group entitled "Flavonoid Derivatives as Potential Cholinesterase Inhibitors in Scopolamine Induced Amnesic Mice; An in vitro, in-vivo and Integrated Computational Approach"  is satisfactory at this point. Authors have responded to the reviewers comment very clearly.

Author Response

Reviewer 2

The manuscript revision submitted by Al-Joufi and group entitled "Flavonoid Derivatives as Potential Cholinesterase Inhibitors in Scopolamine Induced Amnesic Mice; An in vitro, in-vivo and Integrated Computational Approach"  is satisfactory at this point. Authors have responded to the reviewers comment very clearly.

Author response: Worthy reviewer, thanks for accepting and appreciating the comment/suggestion that has been addressed as per valuable suggestions.

Reviewer 3 Report

Major points

  1. The abstract contains too many sentences that are not related to the research results and explanations, so it should be rewritten briefly and clearly.
  2. Statistics need to be improved. In this paper, Dunnett's test was used after ANOVA. However, since this study examined the effects of different doses of F1 ~ F5, Tukey HSD should be performed to clarify the differences between groups treated with different doses.
  3. Need to be rewritten. Lines 477 to 493. Please move to Introduction and rewrite it to avoid redundancy. Discussion should confirm whether the hypothesis of this study has been proven by experiments, and describe the meaning of this study by comparing it with other related research results. Thus, the introduction and the discussion must be rewritten.

Minor revisions

  1. Line 3. Title: change “in-vitro” to “in vitro”.
  2. Line 14 ~ 18. Please fix paragraph alignment error.
  3. Line 48. Other animals also can perform learning and memory.
  4. Line 183, 199 “novel object discrimination (NOD)” to “NOD”
  5. Table 1. R5 is all H, so there is no need to indicate it. Delete R5 and R6. In addition, it is necessary to perform statistical analysis on whether the IC50s of AChE and BChE of F1 to F5 are significantly different. One way ANOVA followed by Tukey HSD post hoc analysis must be performed.
  6. Tables 3 and 4. Statistical differences should be confirmed by performing ANOVA followed by Tukey HSD post hoc analysis.
  7. Figure 5. Low magnification pictures should also be shown. Scale bars should be added to the pictures.
  8. Lines 375, 382, 402, Novel objective Recognition test - Novel objective discrimination (NOD) was used in this manuscript. Please change them.
  9. Table 6. There is typo. “nmol/mg protein)” to “(nmol/mg protein)”. Do Tukey HSD analysis to find differences.

Author Response

Reviewer 3

Major points

Reviewer question #1: The abstract contains too many sentences that are not related to the research results and explanations, so it should be rewritten briefly and clearly.

Author response: Worthy reviewer, thanks for the valuable comment. The abstract has been corrected and rewritten briefly and clearly as per suggestion.

Reviewer question #2: Statistics need to be improved. In this paper, Dunnett's test was used after ANOVA. However, since this study examined the effects of different doses of F1 ~ F5, Tukey HSD should be performed to clarify the differences between groups treated with different doses.

Author response: Worthy reviewer, thanks for the valuable comment. All the data is statistically analyzed with oneway ANOVA followed by Dunnett’s post hoc multiple comparison test to determine the values of P

Reviewer question #3: Need to be rewritten. Lines 477 to 493. Please move to Introduction and rewrite it to avoid redundancy. Discussion should confirm whether the hypothesis of this study has been proven by experiments, and describe the meaning of this study by comparing it with other related research results. Thus, the introduction and the discussion must be rewritten.

Author response: Sir I have checked the discussion superficially. Removed line 482-485 {ref 31-35} as it was repetition.  As per my suggestion the discussion is already as per experimented results. Your experts are required to give a sharp look to this.

Minor revisions

Reviewer question #1: Line 3. Title: change “in-vitro” to “in vitro”.

Author response:  Worthy reviewer, thanks for the valuable comment. The text has been corrected accordingly as per suggestion.

Reviewer question #2: Line 14 ~ 18. Please fix paragraph alignment error.

Author response:  Worthy reviewer, thanks for the valuable comment. The text has been corrected accordingly as per suggestion.

Reviewer question #3: Line 48. Other animals also can perform learning and memory.

Author response:  The 1st paragraph of reference 1 to 2 is specifically focused on cholinergic system with learning and memory associates with cognitive dysfunction. While reference 3 reflects the involvement of cholinergic system in human and animal associates with memory and learning process and methodologies associates with assessment of memory in animals.

Reviewer question #4: Line 183, 199 “novel object discrimination (NOD)” to “NOD”

Author response:  Worthy reviewer, thanks for the valuable comment. The text has been corrected accordingly as per suggestion.

Reviewer question #5: Table 1. R5 is all H, so there is no need to indicate it. Delete R5 and R6. In addition, it is necessary to perform statistical analysis on whether the IC50s of AChE and BChE of F1 to F5 are significantly different. One way ANOVA followed by Tukey HSD post hoc analysis must be performed.

Author response:  Worthy reviewer, thanks for the valuable comment. The text has been corrected in the table accordingly as per suggestion. As far as IC50 is concern, mostly it is presented without any statistical analysis.

Few of the references are given below

  • Amat-ur-Rasool, Hafsa, Fenella Symes, David Tooth, Larissa-Nele Schaffert, Ekramy Elmorsy, Mehboob Ahmed, Shahida Hasnain, and Wayne G. Carter. "Potential nutraceutical properties of leaves from several commonly cultivated plants." Biomolecules 10, no. 11 (2020): 1556.
  • Osmaniye, Derya, Begüm Nurpelin SaÄŸlık, Ulviye Acar Çevik, Serkan Levent, Betül Kaya ÇavuÅŸoÄŸlu, Yusuf Özkay, Zafer Asım Kaplancıklı, and Gülhan Turan. "Synthesis and AChE inhibitory activity of novel thiazolylhydrazone derivatives." Molecules 24, no. 13 (2019): 2392.
  • Kos, Jiri, Tomas Strharsky, Sarka Stepankova, Katarina Svrckova, Michal Oravec, Jan Hosek, Ales Imramovsky, and Josef Jampilek. "Trimethoxycinnamates and Their Cholinesterase Inhibitory Activity." Applied Sciences 11, no. 10 (2021): 4691.
  • Acar Cevik, Ulviye, Begüm Nurpelin Saglik, Serkan Levent, Derya Osmaniye, Betul Kaya CavuÅŸoglu, Yusuf Ozkay, and Zafer Asim Kaplancikli. "Synthesis and AChE-inhibitory activity of new benzimidazole derivatives." Molecules 24, no. 5 (2019): 861.
  • SaÄŸlık, Begüm Nurpelin, Derya Osmaniye, Ulviye Acar Çevik, Serkan Levent, Betül Kaya ÇavuÅŸoÄŸlu, Yusuf Özkay, and Zafer Asım Kaplancıklı. "Design, Synthesis, and Structure–Activity Relationships of Thiazole Analogs as Anticholinesterase Agents for Alzheimer’s Disease." Molecules 25, no. 18 (2020): 4312.

Reviewer question #6: Tables 3 and 4. Statistical differences should be confirmed by performing ANOVA followed by Tukey HSD post hoc analysis.

Author response: Worthy reviewer, thanks for the valuable comment. The Tables 3 and 4 has been corrected and statement regarding statistical analysis has been added as

Mean±SEM (n=3). Statistically analyzed with oneway ANOVA followed by Dunnett’s post hoc multiple comparison test to determine the values of P. *PË‚0.05, **PË‚0.01 and ***PË‚0.001 as comparison of F1 and F5 treated groups  vs. control group using one way ANOVA followed by Dunnett’s comparison.

Reviewer question #7: Figure 5. Low magnification pictures should also be shown. Scale bars should be added to the pictures.

Author response: Worthy reviewer, thanks for the valuable comment. The Figure has been re-modified in manuscript and scale bar along description has been added as

Figure 5. Photomicrograph of kidney and liver from toxicity study of flavonoids. Kidney; (A) control group; (B) group treated with 3000 mg/kg b.w. of F1; (C) group treated with 3000 mg/kg b.w. of F5, Gr: glomeruli, PCT: proximal convoluted tubule, DCT: distal convoluted tubule, P: podocyte. Liver; (D) control group; (E) group treated with 3000 mg/kg b.w. of F1; (F) group treated with 3000 mg/kg b.w. of F5, BD: bile duct, S: sinusoids, EC: endothelial cells, CV: central vein.

Reviewer question #8: Lines 375, 382, 402, Novel objective Recognition test - Novel objective discrimination (NOD) was used in this manuscript. Please change them.

Author response:  Worthy reviewer, thanks for the valuable comment. The text has been corrected in lines 375, 382, 402, accordingly as per suggestion

Reviewer question #8: Table 6. There is typo. “nmol/mg protein)” to “(nmol/mg protein)”. Do Tukey HSD analysis to find differences.

Author response: Worthy reviewer, thanks for the valuable comment. The text has been corrected in Table 6 accordingly as per suggestion

Round 2

Reviewer 1 Report

I have no comments. 

This manuscript is a resubmission of an earlier submission. The following is a list of the peer review reports and author responses from that submission.

Round 1

Reviewer 1 Report

Major points:

1) Authors test IC50 for each compound; how does that in vitro IC50 relate to the chosen dosage used for in vivo studies, i.e., 7.5 mg and 15 mg? Authors need to elaborate on the dosage of compound used for the in vivo studies. How does IC50 inform about the choice of dosage used? If differential potency is found based on in vitro and in silico experiments, how does that relate to the dosage eventually chosen for the study?

  • Authors should include the preliminary data data. 
  •  how does that in vitro IC50 relate to the chosen dosage used for in vivo studies, i.e., 7.5 mg and 15 mg? 

2) Authors chose two dosages for their in vivo studies, i.e., 7.5mg/kg and 15 mg/kg. Though the second dose is double the first dose, neither the behavior assay nor biochemical assays reflect any dosage-dependent changes in the behavior nor the AChE activity /Ach concentration changes? Please clarify. 

  • are there any difference observed while comparing biochemical or behavioral outcomes between 7.5 mg/kg compared to 15 mg/kg dose for any given flavonoid?

3) The discussion requires revision - "how these results from different experiments are interconnected. The authors have to discuss the results in detail. What do changes in any biomarkers report physiologically translate to alteration in amnesia condition? "

4) Conclusion is should be based on the results observed. 

Reviewer 2 Report

This work by Syed Wadood Ali Shah and colleagues evaluated a series of flavone derivatives using in vitro, in-vivo and integrated computational approach for their anticholinesterase potential (against AChE and BuChE). Overall, this is a good study. The writing, presentation and experimental approach are scientifically sound. Discussion and material/ methods sections are well written. There are however some points to consider which I think will improve the understanding and coherence of this manuscript.

Please correct the typos; there are a lot of them in the whole manuscript.

How was the number of animals per lot determined?

How was the dose of 2 mg / kg donepezil chosen? Why didn't you use 10 mg / kg for example?

Has scopolamine been given as a single dose? if so, can a dose of scopolamine even induce cognitive impairment?

Reviewer 3 Report

The manuscript entitled "Flavonoid Derivatives as Potential Cholinesterase Inhibitors in Scopolamine Induced Amnesic Mice; An in vitro, in-vivo and Integrated Computational Approach" by Shah and colleagues is a good attempt to put forward the nootropic effect of the flavone derivatives in scopolamine induced amnesia model. However, the manuscript have some serious flaws which need to be corrected. The experimental design is really not clear and doesn't match very well with the time points claimed. The following are the major concerns from the reviewers:

Major Comments:

1. Why were the doses only 7.5 and 15 mg/kg selected for the study? What was the rationale behind not doing the dose-dependent study?
2. Authors just studied the 600 mg/kg body weight dose for acute toxicity study? What was the reason to just study 600 mg/kg? They should have went as higher as up to 5000 mg/kg (or at least of 3000 mg/kg dose) and then have checked for toxicity if any. 
3. Authors didn't report any toxicity outcomes for e.g. cardiotoxicity, hepatotoxicity, renal toxicity and neurotoxicity data from their compounds. Since this is first time toxicity screening for the flavones derivatives, the vital organs toxicity is of most important to be discussed.
4. Why did authors only selected Y-maze and NOR tests as a paradigm for learning behaviours? Since they are studying the effects of their flavonoids for the first time in memory they should consider all aspects of memory testing for e.g. short term and long term memory (batteries of experiment including Morris water maze, T-maze, Y-maze, active and passive avoidance experiment).
5. Authors mentioned the time for behavioural experiments in between 8am to 12pm. How did they managed to inject the drugs in all groups and then wait for half an hour and then inject scopolamine and carry out experiments. And the behaviour then needed training and test in NOR at an interval of 4 hours? This is not really clear to the reviewer.
6. Were all the compounds tested at once and all the groups were run parallel or the experiments were performed at different times. Were both the Y-maze and NOR performed in the same groups or there were separate groups for the tests. If so how was control managed every time?
7. What was the total number of mice used in the study. The age of mice when the experiments were performed is missing in the manuscript.